# Assessment of the Phytochemical and Nutrimental Composition of Dark Chia Seed (*Salvia hispánica* L.)

**DOI:** 10.3390/foods10123001

**Published:** 2021-12-04

**Authors:** Avilene Rodríguez Lara, María Dolores Mesa-García, Karla Alejandra Damián Medina, Rosa Quirantes Piné, Rafael A. Casuso, Antonio Segura Carretero, Jesús Rodríguez Huertas

**Affiliations:** 1Department of Physiology, Biomedical Research Center, Institute of Nutrition and Food Technology “José Mataix”, University of Granada, Parque Tecnológico de la Salud, Avenida del Conocimiento s/n, 18100 Granada, Spain; avilenerl@correo.ugr.es (A.R.L.); casusopt@gmail.com (R.A.C.); 2Department of Biochemistry and Molecular Biology II, Institute of Nutrition and Food Technology “José Mataix”, Biomedical Research Center, University of Granada, Parque Tecnológico de la Salud, Avenida del Conocimiento s/n, 18100 Granada, Spain; mdmesa@ugr.es; 3Ibs.GRANADA, Biosanitary Research Institute of Granada, 18012 Granada, Spain; 4University Center of Tonala, University of Guadalajara, Av 555 Ejido San José Tateposco, Nuevo Periferico Oriente, Tonala 45425, Mexico; karla.damian@academicos.udg.mx; 5Technological Centre for Research and Development of Functional Foods, Avenida del Conocimiento, 37, 18100 Granada, Spain; rquirantes@cidaf.es (R.Q.P.); ansegura@ugr.es (A.S.C.)

**Keywords:** chia seed, functional food, fiber, omega-3, polyunsaturated fatty acids, essential amino acids, nucleosides, phenolic compounds

## Abstract

Chia seeds are rich sources of different macro and micronutrients associated with health benefits; thus, they may be considered as a functional food. However, the composition depends on the variety, origin, climate and soil. Here, we show a comprehensive characterization of extractable and non-extractable phenolic compounds of dark chia seed *Salvia hispanica* L. using high-performance liquid chromatography–electrospray ionization–quadrupole time-of-flight (HPLC-ESI-QTOF) and discuss potential health benefits associated with the presence of a number of nutritional and bioactive compounds. We report that dark chia from Jalisco is a high-fiber food, containing omega-3 polyunsaturated fatty acids, essential amino acids (phenylalanine and tryptophan), and nucleosides (adenosine, guanidine and uridine), and rich in antioxidant phenolic compounds, mainly caffeic acid metabolites. Our data suggest that chia seeds may be used as ingredients for the development of functional foods and dietary supplements.

## 1. Introduction

Dark chia seed (*Salvia hispánica* L.) is an endemic food of Mexico and Guatemala, and it was used in the pre-Columbian period by Mayan and Aztec civilizations as a staple food combined with amaranth, beans, and corn [1]. Today, chia has reached around 900 species, produced mainly in Mexico, Guatemala, Bolivia, Australia, Peru, Argentina, America and Europe [2], and its seeds are traditionally consumed there and in the southwestern United States. In the European Union countries, the marketing of chia seeds as a new food ingredient has been permitted since 13 October 2009, according to regulation No. 258/97 of the European Parliament and Council [3].

In recent years, there has been particular interest in the nutritional value of chia seeds. Current research studies indicate that this food has many health-promoting properties, for example, on blood lipid profile, as well as hypotensive, hypoglycemic, antimicrobial and on the immune response [4], benefiting non-communicable chronic diseases such as cardiovascular disease and obesity. In this regard, the dietary incorporation of bioactive compounds from chia has become of particular interest [5].

Chia is considered a complete food due to the presence of carbohydrates, mainly fiber, fatty acids, proteins, vitamins, and minerals [6]. In addition, it can be considered a functional food [7], as it is a source of healthy bioactive compounds, mainly antioxidants, with potential applications [8]. The chemical composition of chia seeds has been previously described, highlighting its nutritional value. The elevated fiber content, above some dried fruits, cereals, and nuts, provides metabolic and cardiovascular benefits [9,10].

Omega-3 and omega-6, fatty acids with potential anti-inflammatory effects [11], are also present in chia seed, as well as a good-quality vegetable protein, with essential amino acids [12], and phytochemicals such as a phenolic compound with antioxidant and anti-inflammatory potential [13]. However, the composition depends on the variety, origin, climate, year of cultivation, growing environment and extraction method used [14].

Nowadays, many studies have demonstrated a high biological and technological activity of chia seeds, opening a wealth of potential possibilities for this promising food component [15].

In this regard, a thorough and comprehensive evaluation of the composition of chia seeds is presented in the present work, discussing its relation with health and some technological aspects. Therefore, the aim of the present research is to describe the complete composition of a specific crop of a variety of dark chia from Jalisco (Mexico), describing nutritional and functional effects of these compounds and potential therapeutic uses.

## 2. Materials and Methods

All chemicals were of analytical reagent grade, used as received. Methanol and water LC–MS grade were from Fisher Scientific (Loughborough, UK), and formic acid was from Fluka, Sigma-Aldrich (Steinheim, Germany). All standard compounds were also from Sigma-Aldrich. Dark chia seeds were provided by Mexican producers from Acatic, Jalisco, México.

### 2.1. Proximal Analyses

For the determination of total moisture, 3 g of fresh chia were ground in a homogenization cooled mill Foss Knifetec™ and conducted gravimetric method in a drying oven at 102 °C. After drying, the sample was incinerated in a muffle furnace at 550 °C until they were reduced to white ash. Then, samples were quantified by gravimetry. Total proteins were obtained by the Kjeldhal method after sulfuric acid digestion (AOAC 1997) [16]. The amount of total fat was determined following the Soxleht method. Briefly, 2 g of sample were hydrolyzed with hydrochloric acid 4 M and extracted with petroleum ether and dried at 40–60 °C. Total fat was calculated by gravimetry after extraction (AOAC 1997). The carbohydrate content was estimated as a nitrogen-free extract (NFE) by the difference from the sum of the protein, fat, ash and crude fiber content [17]. The determination of total dietary fiber was performed by the enzymatic gravimetric method (AOAC 1997).

### 2.2. Fatty Acid Profile Analysis

The fatty acid composition was determined by gas chromatography following the direct transesterification method of rust from 100 mg of ground sample based on a modification of the Lepage and Roy method [18]. Fatty acids were identified by comparing with an external standard (37 Component FAME Mix Ref. 47885-U. Sigma-Aldrich). The quantification of each fatty acid was performed by extrapolation in calibration curves, and data were expressed as percentages of the total amount. Gas Chromatography–Mass Spectometry (GC–MS) was performed in an Agilent A7890 (Agilent Technologies, Palo Alto, California, USA). The initial temperature was 140 °C (2 min)-temperature ramp 4 °C/minute-maximum temperature 310 °C (6 min), injector temperature (250 °C), interface and source temperature (240 °C), capillary column SP 25–60 (100 × 0.25 × 0.25); 100 m length, 0.25 internal diameter and 0.25 microns film thickness. The mass conditions were full scan 45–500 Dalton, Electron impact ionization, El+ at 70 e V Quattro micro-GC, waters (Manchester, UK).

Lipid quality indices were calculated according to Ulbricht and Southgate (1991) [19]. The atherogenic index (AI) was calculated as follows:AI = [12 : 0 + 4 (14 : 0 + 16 : 0)]
[(n6 + n3) PUFA + 18 : 1 + ΣMUFA]

The index of thrombogenicity (IT) was calculated as follows:IT = (14 : 0 + 16 : 0 + 18 : 0)
[(0.5 × 18 : 1) + 0.5 (ΣMUFA) + 0.5 (n6PUFA) + 3 (n3PUFA) + (n3PUFA/n)

### 2.3. Phenolic Compounds Determination

#### Sample Preparation

Chia seeds were ground in an Ultra Centrifugal Mill ZM 200 (Retsch, Haan, Germany) at 10,000 rpm, and passed through a 0.1 mm ring sieve. Phenolic compounds from chia seeds were extracted according to Martínez-Cruz et al. (2014) [20] with slight modifications. In brief, 0.5 g of ground seeds was extracted in triplicate with 10 mL aqueous methanol 70% in a Branson 3510 ultrasonic bath (Branson Ultrasonic Corp., St. Louis, MO, USA) operating at 25 °C and 40 kHz for 15 min. Then, the mixture was centrifuged at 7500 rpm for 10 min at room temperature in a Sorvall ST 16R centrifuge (Thermo Fisher Scientific, Leicestershire, UK). The supernatant was evaporated in a rotary evaporator (BUCHI Iberica S.L.U., Barcelona, Spain) at 40 °C, reconstituted with 70% methanol at a concentration of 5 mg/mL and filtered through a 0.22 µm cellulose syringe filter before injection into the chromatographic system. To perform the quantitative analyses, gallic acid at a concentration of 10 μg/mL was added to each extract as internal standard.

The main compounds found in the extract were quantified whenever standards were available, although some of them were tentatively quantified on the basis of other compounds having similar structures.

### 2.4. HPLC–ESI–QTOF–MS Analysis

The HPLC analyses were performed on an Agilent 1260 HPLC instrument (Agilent Technologies, Palo Alto, CA, USA) equipped with a binary pump, an online degasser, an auto-sampler, a thermostatically controlled column compartment, as well as a diode array detector. The samples were separated on an Agilent ZorBax Eclipse Plus C18 column (1.8 µm, 4.6 × 150 mm). The mobile phases consisted of water with 0.1% formic acid (A) and methanol (B) using a gradient elution according to the following profile: 0 min, 5% B; 15 min, 30% B; 20 min, 95% B; 25 min, 5% B. The initial conditions were maintained for 5 min. The flow rate was 0.5 mL/min, the column temperature, 25 °C, and the injection volume, 5 µL.

Detection was performed using an Agilent 6540 Ultra-High-Definition (UHD) Accurate Mass Q-TOF mass spectrometer within a mass range of 50–1700 m/z, operating in negative ion mode and equipped with a Jet Stream dual ESI interface since this ionization mode is more appropriate in terms of compounds ionization. The operating parameters were as follows: drying gas flow rate, 10 L/min; drying gas temperature, 325 °C; nebulizer, 20 psi; capillary, 4000 V; fragmentor, 130 V. The MS/MS analyses were acquired by automatic fragmentation where the two most intense mass peaks were fragmented. Collision energy values for MS/MS experiments were adjusted at 10, 20 and 40 eV. Nitrogen was used as drying, nebulizing and collision gas. Continuous infusion of the reference ions m/z 112.985587 (trifluoroacetate anion) and 1033.988109 (adduct of hexanes (1H,1H, 3H-tetrafluoropropoxy) phosphazine or HP921 was used to correct each mass spectrum.

All the operations, acquisition and analysis of data were controlled by Masshunter workstation software version B.06.00 (Agilent Technologies, USA).

Calibration graphs of seven points in triplicate were obtained with a set of solutions of sucrose, citric acid, tyrosine, phenylalanine, protocatechuic acid, gentisic acid, vanillic acid, caffeic acid, ferulic acid, rosmarinic acid and luteolin were prepared using gallic acid at a concentration of 10 μg/mL as internal standard.

Table 1 shows the calibration data including limits of detection (LODs) and limits of the quantification of specific (LOQs) for individual compounds in standard solutions, which were calculated as S/N = 3 and S/N = 10, respectively, where S/N is the signal-to-noise ratio. The compound concentrations were determined using the corrected area of each individual compound (three replicates) and by interpolation in the corresponding calibration curve.

## 3. Results

### Characterization of Chia Seeds 

Table 2 presents the composition of chia seeds, and Table 3 describes the composition of the fatty acids found in the black variety of chia seeds.

Figure 1 shows the base peak chromatogram (BPC) in negative ion mode of dark chia seeds extract. The main compounds have been numbered according to their retention times. These compounds were tentatively identified whenever possible by interpretation of their MS and MS/MS spectra obtained by QTOF–MS combined with the data provided by databases (SciFinder, PubChem and Mass bank) and the literature.

Table 4 summarizes the MS data of the identified compounds found in the chia seeds extract, including the proposed molecular formula and detected species, experimental m/z, m/z error and score of the selected molecular formula, and the main fragments obtained by MS/MS, as well as the proposed compound for each peak and the references used for their identification.

Table 5 summarizes the quantitative results obtained for the extract; the compound concentrations were determined using the corrected area of each individual compound (three replicates) and by interpolation in the corresponding calibration curve. Sucrose, citric acid, tyrosine and rosmarinic acid were quantified by using their corresponding standard curve since their standards were available, while the rest of the compounds were tentatively quantified on the basis of the other standards having similar structures, as indicated in Table 5. It should be taken into account that the response of the standards can differ from that of the analytes present in the extract, and consequently, the quantification of these compunds is only an estimation of their actual concentrations. In any case, they can be considered to be a useful approximation.

## 4. Discussion

Chia seeds are considered a good source of different components, particularly due to mainly the crop, and environmental and agronomic factors. In this work, a phytochemical analysis of the components of dark chia seed *Salvia hispanica* L. from Mexico was carried out, with the aim of understanding its composition, and its potential benefits for human health and possible nutraceutical properties. Our results describe that the main components with nutritional and functional activities are fiber, linolenic acid, essential amino acids, nucleosides and phenolic compounds, reinforcing the concept of chia as a complete and functional food [47].

Chia seeds are considered to have high nutritional value, particularly thanks to their high content of dietary fibers. It ranges between 30.2% [48] and 34.4% [49]. The present data show that dark chia seeds from Jalisco contain 35.1% fiber, which is slightly higher than the 32.6% reported for a dark chia variety from Sinaloa [50]. Fiber composition not only depends on the variety [7], but also on other environmental factors [51]. The multiple functions and benefits of dietary fiber intake include the reduction of coronary heart disease risk, and the reduction of glycaemic index. It reduces the risk of type 2 diabetes and some types of cancer, and it also may exert a high satiety effect that may help control obesity [52]. These actions justify the recommended daily intakes (RDI): 25–30 g/d. [53]. According to the Royal Decree 1334/1999, to be considered a “high fiber” food, the product must contain at least 6 g of fibre per 100 g of food, or at least 3 g of fibre per 100 calories. Therefore, including high-fiber foods such as chia is a good choice in order to reach these recommendations [54].

Furthermore, fiber may have different applications in the food industry. Fiber gums are extracted from chia seeds and are used as an additive to control the viscosity, stability, texture and consistency of elaborated foods [55]. Other authors have described that chia mucilage contains polysaccharides, which are currently used for the functional coating of some foods, replacing synthetic packaging and contributing to environmental sustainability [56].

Other important carbohydrates found naturally in the raw chia are oligosaccharides, which can be used as high-energy molecules for the body, but also may exert benefits for health. Within them, stachyose (tetrasaccharide), and two isomers of raffinose (trisaccharides) have been found in these dark chia seeds. These oligosacharides are commonly used as a sugar substitute, in order to decrese the glycaemic index of manufactured foods containing chia seeds [57]. In addition, it has been described that trisaccharides may contribute to the formation of natural antioxidant phenols, suggesting another potential benefit for these chia seed components [58].

As previously reported, dark chia seeds contain around 86% of unsaturated fats, mainly 59% of linolenic acid, 19% of linoleic acid and 8% of oleic acid [59,60,61,62]. This proportion of linolenic acid is considerably higher than those present in other cereals such as rice (2.1%), maize (1%), wheat (0.08), quinoa (6.7%) and amaranth (1.01%), and makes chia the best source of linolenic acid as the precursor of long-chain omega-3 polyunsaturated fatty acids [63,64,65,66] However, these fatty acid compositions can vary according enviromental factors, such as geographical location, temperature changes, the composition of the soil, within others. For example, the content of unsaturated ω-3 fatty acids increases with lower temperatures [67,68], and can reach up to 68% of ω-3 linolenic acid [69]. Linolenic acid has demonstrated beneficial effects, since it is the precursor of long-chain polyunsaturated fatty acids, eicosapentaenoic and docosahexaenoic acid, two anti-inflammatory fatty acids which may modulate the immune system behaviour and exert cardioprotective [70], anti-diabetic [71], hepatoprotective [72] and anti-cancer effects. [67]. The omega 6/omega 3 ratio in chia seeds showed a value of 3.02. Finally, the atherogenic and thrombogenic index showed a low value, indicating high amounts of fatty acids with antiatherogenic properties in the chia seed extract [73]. Therefore, the inclusion of chia seeds in the diet would decrease the ω-6:ω-3 ratio, contributing to a balanced diet and anti-inflammatory benefits.

Environmental and agronomic factors also influence protein content [74]. Chia seeds are considered a good source of vegetable protein that accounts for 18–24% of the total mass in the dark variety [49,50,51,52,53,54,55,56,57,58,59,60,61,62,63,64,65,66,67,68,69,70,71,72,73,74,75], which is consistent with the amount reported in the present work, and higher than common cereals and oilseeds, such as wheat (14%), barley (9.2%) [76], oats (15.3%), maize (14%), and rice (8.5%) [77]. In addition, the quality of this protein has been reported as high [78]. High-quality protein can be determined by the amount of protein in the nourishment, the amount of essential amino acids and the digestibility of the protein [79].

Our analysis of the specific amino acid confirms the presence of five main amino acids: L-asparagine, L-aspartic acid, L-tyrosine, L-phenylalanine and L-tryptophan, as described by [80]. Within them, tryptophan and phenylalanine are two essential amino acids that are scarce in foods. They play an important role in the organism. Tryptophan is a precursor of serotonin and melatonin involved in the regulation of circadian rhythms, as well as in the regulation of appetite and mood; while phenylalanine is a precursor of dopamine, a hormone that generates well-being, pleasure, euphoria and happiness [11]). The amount of phenylalanine in chia seeds (around 1.02 g/100 g) makes this cereal a good source of this essential amino acid, similar to almonds (1.13 g), pistachios (1.09 g), flax seeds (0.96 g), sesame seeds (0.94 g), and pumpkin seeds (0.92 g). Other seeds provide less amounts, such as walnuts (0.71 g); grains, such as oats (0.89 g), rye (0.43 g) and wheat (0.46 g); and legumes, such as beans (0.47 g), soybeans (0.56 g) and lentils (0.44 g), while only hemp seeds provide higher amounts (2.77 g), [81,82,83]. Therefore, we can say that the presence of these two essential amino acids in chia seeds can be a good source of quality dietary protein, as described above [22]. In addition, the absence of gluten makes chia seeds highly valued for patients suffering from celiac disease. Furthermore, isolated chia seeds proteins may be a useful ingredient that can be potentially added to different enriched-manufactured foods [84].

To our best knowledge, this is the first work that has described the presence of nucleosides, adenosine, guanosine and uridine in chia seeds, providing a new possible nutritional advantage that needs to be investigated. The intake of these molecules has demonstrated health benefits with effects on the prevention of different pathologies and immunological protection, mainly in infants and fragile adults. These benefits have led to their incorporation in different types of foods, such as infant formulas [85], enteral formulas for adults [86], and food supplements [87,88]. Various interventional studies have been conducted to substantiate the functionality of nucleotides, and have linked them to the stimulation and modulation of the immune system [89]. They are involved in the proliferation, maturation and activation of lymphocytes, in the stimulation of the phagocytic function of macrophages, and in the modulation of responses to grafts, tumours, infections, delayed hypersensitivity and the production of immunoglobulins [90]. On the other hand, the EFSA has stated that nucleotides are conditionally essential nutrients, since they contribute to the development and growth of new cells, promoting general well-being, enhancing body immunity and helping to promote an adequate bowel activity [91]. Therefore, the inclusion of chia in the diet may help to reach these specific recommendations, and new clinical studies are necessary in order to ascertain potential helth benefits of chia nucleosides supplemmentation.

Focusing on the phenolic content, and according to our results, the following phenols were identified: salvianolic acid F, two dihydroxybenzaldehyde isomers, methyl rosmarinate, rosmarinic acid, rosmarinic acid glucoside, salvianolic acid B/E, dihydroxybenzoic acid, fertaric acid, caffeoyl glucoside, caffeoyl gentiobiose, sibiricose A3, dihydroxybenzoic acid apiosyl glucoside, caftaric acid, oresbiusin A, and in lower concentration: vanillic acid glucoside, salvianic acid, dihydroxybenzoyl glucoside and dihydroxyphenylacetic acid. Most of them have been described previously [23].

Our data show specifically that rosmarinic acid is one of the most abundant compounds in black chia seeds. Rosmarinic acid is a caffeic acid derivative and has been reported as one of the main phenolic compounds present in chia seeds. Martinez-Cruz and Paredes-Lopez et al. (2014) [20] reported (0.92 mg/g). Our data showed a concentration of 3.9 mg/g, specifically rosmarinic acid glucoside and 1.2 mg/g of rosmarinic acid. The presence of this acid is of utmost importance, because it is involved in biological activities as it has antioxidant, astringent, anti-inflammatory, anti-thrombotic, anti-mutagenic, antibacterial and antiviral properties [92,93,94].

On the other hand, to date, there have been no reports describing salvianolic acids (a dimer of caffecic acid) in chia seeds, and the absence of caffecic acid. Other works have observed the presence of trimers and tetramers of the caffeic acid. This molecule is the building block of a variety of plant metabolites make-up from multiple condensation products, giving rise to a variety of oligomers including salvianolic acid A-K and lithospermic acid, in other Salvia species such as S. miltiorhiza. S. Officinalis, S. cavaleriei, S. flava, S chinensis [95,96,97], and identified salt A and salt B as the most abundant compounds among the salvianolic acids [98,99]. However, our results show the presence of salvianolic acid, salvianolic acid B/E and salvianolic acid F specifically in the dark chia seeds from Jalisco. Among the beneficial properties of these compounds, we highlight anti-inflammatory, antioxidant and free radical scavenging effects [100]. As other phenolic compounds, they exert strong antioxidant activity and other biological activities. The antioxidant activity of salvianolic acids is mediated by the increase of the expression of antioxidant enzymes, while decreased expression of pro-antioxidant enzymes [99]. Furthermore, recent studies show that salvianolic acids have good effects on some chronic fibrotic diseases, especially liver and pulmonary fibrosis, and it also has possible anticancer effects. Other potential functions and possible mechanisms of action are currently being studied to be better applied for the treatment of other diseases [101].

Finally, apart from the nutritional and functional benefits, chia seeds do not contain other components that may exert harmful effects in specific physiological or pathological situations, such as gluten or other allergens, or micotoxins [102,103,104], enabling their inclusion as an ingredient/complement into different foods [105].

## 5. Conclusions

There has been a great deal of interest in the nutritional benefits of chia seeds. In general, chia seeds are rich in dietary fiber and oligosaccharides, omega-3 polyunsaturated linolenic acids, high-quality protein containing L-phenylalanine and L-tryptophan. They are also a good source of a number of bioactive compounds with high antioxidant activity, in particular, polyphenols. In addition, the specific variety of dark chia from Jalisco contains as well nucleosides: adenosine, guanosine and uridine, improving its nutritional value. Our findings indicate that chia seeds have many health-promoting properties and may be considered as a complete functional food ingredient apart from their technological properties.

## Figures and Tables

**Figure 1 foods-10-03001-f001:**
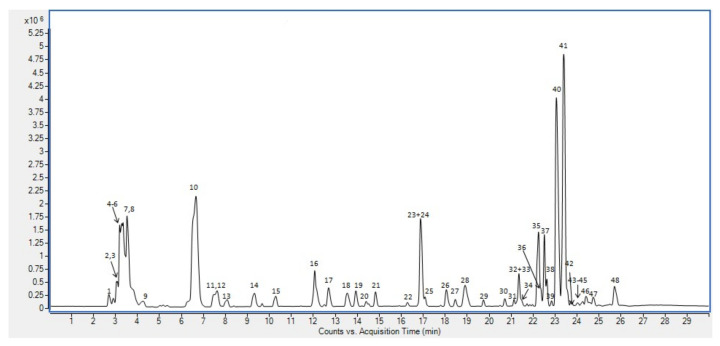
Base peak chromatogram of chia seeds extract. The peaks are identified with numbers according to their order of elution.

**Table 1 foods-10-03001-t001:** Calibration data used for the quantification of specific compounds.

Analyte	Calibration Equation	Calibration Range (μg/mL)	*R^2^*	LOD (μg/mL)	LOQ (μg/mL)
Sucrose	y = 0.1602x + 1.0751	10–50	0.9752	0.05 ± 0.01	0.18 ± 0.05
Citric acid	y = 0.012x + 0.194	10–50	0.9701	0.9 ± 0.3	3.0 ± 0.9
Tyrosine	y = 0.1248x + 0.0364	1–10	0.9906	0.071 ± 0.005	0.24 ± 0.02
Phenylalanine	y = 0.0634x + 0.0062	1–10	0.9886	0.10 ± 0.01	0.32 ± 0.05
Protocatechuic acid	y = 0.0264x − 0.0201	1–10	0.9699	0.31 ± 0.06	1.0 ± 0.2
Gentisic acid	y = 0.0390x − 0.0050	1–10	0.9843	0.19 ± 0.03	0.6 ± 0.1
Vanillic acid	y = 0.0293x + 0.0081	1–10	0.9877	0.18 ± 0.03	0.59 ± 0.08
Caffeic acid	y = 0.2119x − 0.0094	0.25–5	0.9864	0.027 ± 0.004	0.09 ± 0.01
Ferulic acid	y = 0.1207x + 0.0135	0.5–5	0.9759	0.041 ± 0.004	0.14 ± 0.01
Rosmarinic acid	y = 0.1067x + 0.0398	0.5–20	0.9827	0.028 ± 0.005	0.09 ± 0.02
Luteolin	y = 2.0691x + 0.0586	0.05–1	0.9933	0.0019 ± 0.0004	0.006 ± 0.001

LOD: limit of detection; LOQ: limit of quantification.

**Table 2 foods-10-03001-t002:** Proximal composition of chia seeds in 100 g.

Component	Composition
Carbohydrates	21.8%
Sugars	<0.1%
Fiber	35.1%
Fat	18.3%
Saturated fat	1.2%
Insaturated fats	17.1%
Protein	18.8%
Ash	4.7%
Moisture	5.8%
Energy (kcal)	388.3
Energy (kJ)	1612

**Table 3 foods-10-03001-t003:** Determination of fatty acid composition of chia seeds.

Fatty Acids	Composition (%)
Miristic C14:0	0.41
Palmític C16:0	7.49
Palmitoleic C16:1 cis-9	0.5
Stearic C18:0	4.0
Oleic C18:1 cis-9	8.34
Linoleic C18:2 cis-9,12 (ω-6)	19.5
α-Linolenic C18:3 cis-6,9,15 (ω-3)	59.2
ƴ-Linolenic C18:3 cis-6,9,12 (ω-6)	0.52
W6:w3	3.02
Atherogenic index	0.045
Thrombogenic index	0.007

**Table 4 foods-10-03001-t004:** Compounds detected in chia seeds extract. (Figure 1) classified according the chemical nature of the compound.

Peak	RT (min)	Formula	Species	m/z	Error (ppm)	Score	MS/MS Spectra Peaks (Relative Intensity)	Proposed Compound	Chemical Nature	Reference
1	2.701	-	-	272.9574	-	-	158.9765 (100), 114.9868 (16.55)	Unknown	-	-
2	3.019	C_4_H_8_N_2_O_3_	(M-H)^−^	131.0463	−0.52	99.95	-	L-Asparagine	Amino acid	[21]
3	3.091	C_4_H_7_NO_4_	(M-H)^−^	132.0298	2.98	98.97	115.0025 (100)	L-Aspartic acid	Amino acid	[22,23]
4	3.186	C_24_H_42_O_21_	(M-H) (M+Cl) (M+COOH)^−^	665.2153 701.1920 711.2192	−0.67	98.99	179.0543 (100), 665.2119 (41.13), 101.0231 (35.57), 119.0332 (25.79), 221.0643 (23.52), 161.0437 (21.31)	Stachyose	Sugar	[24]
5	3.281	C_18_H_32_O_16_	(M-H) (M+Cl) (M+COOH)^−^	503.1642 539.1411 549.1699	−4.51	89.54	503.1593 (100), 539.1364 (50.51), 179.0544 (14.76), 119.0335 (4.15)	Raffinose isomer 1	Sugar	[23,24]
6	3.351	C_25_H_38_O_22_	(M-H)^−^	689.1802	−2.5	91.35	185.0206 (100)	Unknown	-	-
7	3.511	C_12_H_22_O_11_	(M-H)^−^ (M+COOH)^-^	341.1086 387.1140	1.29	98.13	-	Sucrose *	Sugar	[23,25]
8	3.565	C_18_H_32_O_16_	(M-H)^−^ (M+Cl)^−^ (M+COOH)^-^	503.1628 539.1383 549.1685	−0.18	99.46	209.0672 (100), 503.1656 (85.25), 179.0566 (19.31)	Raffinose isomer 2	Sugar	[23,24]
9	4.22	C_4_H_6_O_5_	(M-H)^−^	133.0145	−1.98	99.52	115.0026 (100), 107.0357 (43.12), 133.0109 (24.5)	Malic acid	Organic acid	[25]
10	6.623	C_6_H_8_O_7_	(M-H)^−^	191.0196	0.71	99.52	111.0062 (100), 191.0153 (28.29), 129.0161 (8.26)	Citric acid *	Organic acid	[5,23,24,25]
11	7.458	C_9_H_12_N_2_O_6_	(M-H)^−^ (M+Cl)^−^ (M+COOH)^-^	243.0622 279.0383 289.0672	0.41	99.87	110.0246 (100), 200.0572 (57.92), 243.0623 (55.74), 140.0357 (23.12), 152.0348 (21.31)	Uridine	Pyrimidine nucleoside	[26]
12	7.677	C_23_H_32_O_19_	(M-H)^−^	611.1443	4.98	73.97	306.0776 (100), 611.1493 (65.81), 272.0894 (26.19), 338.0497 (12.97)	Unknown	-	-
13	8.06	C_9_H_11_NO_3_	(M-H)^−^	180.0667	−0.35	99.65	119.0507 (100), 163.0405 (30.56)	L-Tyrosine *	Amino acid	[22,27,28]
14	9.307	C_10_H_13_N_5_O_4_	(M-H)^−^ (M+Cl)^−^ (M+COOH)^−^	266.0890 302.0656 312.0949	0.2	99.5	134.0473 (100), 266.0915 (13.86)	Adenosine	Pyrimidine nucleoside	[26]
15	10.279	C_10_H_13_N_5_O_5_	(M-H)^−^	282.0849	−1.86	97.24	133.016 (100), 108.0206 (93.79), 150.0424 (32.97)	Guanosine	Pyrimidine nucleoside	[26]
16	12.063	C_42_H_51_NO_17_	(M-H)^−^	840.3096	−0.65	90.76	840.3184 (100), 241.004 (5.54)	Unknown	-	-
17	12.7	C_9_H_11_NO_2_	(M-H)^−^	164.0716	0.9	99.68	103.0551 (100), 147.0452 (34.09)	L-Phenylalanine *	Amino acid	[22,27,29,30]
18	13.549	C_8_H_8_O_4_	(M-H)^−^	167.0346	2.6	98.72	123.0448 (100), 121.0294 (73.95), 109.0291 (33.23), 137.0243 (23.82)	Dihydroxyphenylacetic acid	Phenolic acid	[31]
19	13.937	C_13_H_16_O_9_	(M-H)^−^	315.0714	2.59	97.69	152.0116 (100), 108.0215 (70.99), 153.0191 (52.14), 109.0294 (37.98)	Dihydroxybenzoyl glucoside	Phenolic acid	[32]
20	14.407	C_9_H_10_O_5_	(M-H)^−^	197.0448	3.87	96.9	179.0359 (100), 135.0456 (57.62), 123.0456 (55.32)	Salvianic acid A	Phenolic acid	[5,30]
21	14.829	C_14_H_18_O_9_	(M-H)^−^ (M+Cl)^−^	329.0888 365.0649	−2.65	96.17	167.0355 (100), 123.0455 (5.58)	Vanillic acid glucoside	Phenolic acid	[32]
22	16.284	C_10_H_12_O_5_	(M-H)^−^	211.0614	−0.9	99.34	181.0509 (100), 163.0397 (57.5)	Oresbiusin A	Catechol	[33]
23	16.887	C_57_H_59_N_5_O_21_	(M-H)^−^	1148.363	0.33	98.46	822.305 (100), 1048.3322 (90.14), 241.0041 (4.03)	Unknown	-	-
24	16.902	C_13_H_12_O_9_	(M-H)^−^	311.0393	5.38	90.4	149.0086 (100), 179.0345 (98.75), 135.0442 (10.77)	Caftaric acid	Phenolic acid	[34,35]
25	17.082	C_18_H_24_O_13_	(M-H)^−^	447.1142	0.63	99.01	152.0115 (100), 108.0214 (38.37), 109.0292 (16.72)	Dihydroxybenzoic acid apiosyl glucoside	Phenolic acid	-
26	18.051	C_11_H_12_N_2_O_2_	(M-H)^−^	203.0828	−0.77	99.54	116.0502 (100), 142.0657 (26.79)	L-Tryptophan	Amino acid	[28,29,36]
27	18.46	C_19_H_26_O_13_	(M-H)^−^	461.1291	2.21	97.26	137.0248 (100), 461.133 (57.96), 239.0571 (22.15)	Sibiricose A3	Phenolic acid	[37]
28	18.911	C_7_H_6_O_3_	(M-H)^−^	137.0247	−1.61	99.53	108.0213 (100)	Dihydroxybenzaldehyde isomer 1	Phenol	[38]
29	19.74	C_17_H_22_O_11_	(M-H)^−^	401.1081	2.18	97.34	267.0734 (100), 401.1126 (95.86), 249.0633 (69.13), 151.0407 (65.01), 113.0246 (46.11)	Unknown	-	-
30	20.713	C_21_H_28_O_14_	(M-H)^−^ (M+Cl)^−^	503.1404 539.1169	0.56	99.53	503.1465 (100), 161.0255 (33.26), 323.08 (13.1), 281.068 (11.53)	Caffeoyl gentiobiose	Phenolic acid	[39]
31	21.13	C_20_H_34_O_11_	(M-H)^−^	449.202	1.98	98.03	112.9853 (100), 449.2049 (39.63), 248.9621 (46.98), 167.1072 (37.07), 180.9744 (29.26)	Unknown	-	-
32	21.279	C_15_H_18_O_9_	(M-H)^−^	341.087	2.46	97.68	179.0355 (100), 135.0451 (24.28)	Caffeoyl glucoside	Phenolic acid	[33]
33	21.353	C_14_H_14_O_9_	(M-H)^−^	325.0545	6.48	86.2	-	Fertaric acid	Phenolic acid	[40]
34	21.458	C_7_H_6_O_4_	(M-H)^−^	153.0192	1.15	99.74	109.0294 (100), 135.0087 (21.35)	Dihydroxybenzoic acid	Phenolic acid	[41]
35	22.244	C_18_H_28_O_9_	(M-H)^−^	387.1677	−4.14	92.9	101.0244 (100), 113.0246 (39.64), 163.1133 (20.55), 119.0346 (17), 207.1031 (11.35)	Tuberonic acid glucoside	Iridoid	[42]
36	22.347	C_48_H_65_N_3_O_28_	(M-H)^−^	1130.368	0.96	95.82	1130.3801 (100), 164.9867 (8.73)	Unknown	-	-
37	22.509	C_48_H_65_N_3_O_28_	(M-H)^−^	1130.3675	0.83	97.95	-	Unknown	-	-
38	22.623	C_36_H_30_O_16_	(M-H)^−^	717.1457	0.84	98.31	519.0981 (100); 475.1072 (62.91), 339.0535 (36.96)	Salvianolic acid B/E	Phenolic acid	[40]
39	22.84	C_36_H_27_NO_12_	(M-H)^−^	664.1444	3.15	81.38	272.09 (100), 664.1502 (98.4), 502.1162 (83.26), 391.0527 (61.95), 229.0183 (29.74)	Unknown	-	-
40	23.075	C_24_H_26_O_13_	(M-H)^−^	521.1293	3.33	84.3	359.0653 (100), 323.0659 (91.9), 161.0182 (80.26), 197.0381 (22.23), 179.0282 (20.33)	Rosmarinic acid glucoside	Phenolic acid	[5]
41	23.394	C_18_H_16_O_8_	(M-H)^−^	359.0772	1.84	89.16	161.0189 (100), 197.039 (37.86), 179.0287 (15.9)	Rosmarinic acid *	Phenolic acid	[5,23,34]
42	23.545	C_20_H_20_O_7_	(M-H)^−^	371.1131	1.69	98.6	177.0562 (100), 162.033 (31.34), 193.0513 (22.61)	Salviandulin B/Tehuanin G/C/F /E	Iridoid	[43]
43	23.732	C_19_H_18_O_8_	(M-H)^−^	373.092	2.49	97.29	197.0464 (100), 135.0456 (96.83), 175.0402 (95.03), 179.0355 (33.23)	Methyl rosmarinate	Phenolic acid	[5,40]
44	23.757	C_20_H_18_O_5_	(M-H)^−^	337.1074	2.34	97.93	307.0632 (100), 322.0868 (47.61), 279.0662 (26.64)	Methyltanshinonate	Other compounds	[44]
45	24.252	C_7_H_6_O_3_	(M-H)^−^	137.0243	1.02	99.6	137.0243 (100), 109.0634 (1.37)	Dihydroxybenzaldehyde isomer 2	Phenol	[38]
46	24.411	C_17_H_14_O_6_	(M-H)^−^	313.0706	3.92	94.39	161.0245 (100), 295.2296 (28.77), 313.0733 (9.18), 151.04 (7.05)	Salvianolic acid F	Phenolic acid	[45]
47	24.74	C_17_H_14_O_7_	(M-H)^−^	329.0652	4.74	92.22	299.02 (100), 314.0432 (74.91)	Jaceosidin	Flavonoid	[46]
48	25.718	C_21_H_32_O_7_	(M-H)^−^	395.2091	−3.16	88.6	-	Unknown	-	-

The compounds identified in the chia seed extracts are described. Retention time (RT) and consecutive peak number corresponding to Figure 1 are indicated. The proposed molecular formula, experimental m/z, error (ppm), score, MS/MS spectrum peaks (relative intensity), proposed compound and chemical nature. * Identification confirmed using commercial standards.

**Table 5 foods-10-03001-t005:** Concentration of the main compounds in chia seeds Value = X ± SD.

Peak	Compound	Calibration Curve Used	Concentration Mean ± SD (µg/g Seeds)
7	Sucrose	Sucrose	5.1 × 10^3^ ± 0.8 × 10^3^
10	Citric acid	Citric acid	503.9 ± 82.1
13	L-Tyrosine	Tyrosine	33.8 ± 1.7
17	L-Phenylalanine	Phenylalanine	51.0 ± 11.9
18	Dihydroxyphenylacetic acid	Protocatechuic acid	22.8 ± 0.004.7
19	Dihydroxybenzoyl glucoside	Gentisic acid	189.6 ± 32.4
20	Salvianic acid A	Protocatechuic acid	40.1 ± 6.7
21	Vanillic acid glucoside	Vanillic acid	165.7 ± 29.1
22	Oresbiusin A	Protocatechuic acid	15.0 ± 2.5
24	Caftaric acid	Caffeic acid	7.5 ± 2.4
25	Dihydroxybenzoic acidapiosyl glucoside	Gentisic acid	80.5 ± 6.1
28	Dihydroxybenzaldehyde	Gentisic acid	56.3 ± 2.1
30	Caffeoyl gentiobiose	Caffeic acid	4.0 ± 0.4
32	Caffeoyl glucoside	Caffeic acid	9.5 ± 1.7
33	Fertaric acid	Ferulic acid	15.8 ± 3.7
34	Dihydroxybenzoic acid	Gentisic acid	<LOQ *
38	Salvianolic acid B/E	Rosmarinic acid	61.6 ± 13.5
40	Rosmarinic acid glucoside	Rosmarinic acid	3.9 × 10^3^ ± 0.7 × 10^3^
41	Rosmarinic acid	Rosmarinic acid	1.2 × 10^3^ ± 0.1 × 10^3^
43	Methyl rosmarinate	Rosmarinic acid	13.5 ± 0.5
47	Jaceosidin	Luteolin	0.7 ± 0.1

Data are expressed as mean ± standard deviation. * LOQ: limit of quantification.

## Data Availability

Not applicable.

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
