# Peer review of "Assessment of the Phytochemical and Nutrimental Composition of Dark Chia Seed (Salvia hispánica L.)"

_foods, 2021, doi:10.3390/foods10123001_

Round 1

Reviewer 1 Report

The manuscript titled “Assessment of the phytochemical and nutrimental composition of dark Chia seed (Salvia Hispánica L.)” is related to the evaluation of the chemical composition of chia seeds from Mexico. The chemical composition of chia seeds was already studied which is a drawback of the manuscript. What was done should be presented in the introduction which is currently missing. However, applied analytical methods are appropriate and conclusions are supported by the results. There as several major and minor issues listed below which should be addressed.

Major issues:

Introduction – chemical composition of the chia seed was already extensively analysed in the previous studies. Authors should write a mini-review and address what was already achieved in the field. Also, authors should emphasize the novelty of the proposed manuscript to justify its necessity.

English should be polished by a native speaker, some of the sentences are hard to follow.

Minor issues:

The numbering of section 2 should be corrected as 2.2 FA 2.3 Phenolic compounds determination etc.

Section 2.2 – Like for HPLC MSMS method, please add a brief description of sample preparation, GC instrumental setup (time of analysis, temperature programme, column, gas flow etc) and compounds identification procedure.

Lines 87-88 – Please provide information about ultrasonic bath (model, producer etc), frequency and temperature of extraction. Also infr0omation about centrifuge and rotary evaporator

Lines 107-108 – Please add an explanation why only negative mode was performed

Lines 118-120 – Calibration graphs were obtained with a set of solutions of sucrose, citric acid etc and gallic acid as internal standard. Please rephrase the sentence.

Table 4  - “*” should be explained below the table. I believe that malic acid was appointed with “*” by mistake, please correct this. A more suitable title would be “Compounds detected in chia…” since not all compounds were identified.

Line 178 – “improvement of type 2 diabetes and some types of cancer” fibers do not improve disease but rather reduce the risk of their manifestation, please rephrase the entire sentence.

Lines 200-211 Since the manuscript is oriented towards also to functional properties and fatty acid profile please calculate ω6/ω3 ratio, atherogenicity index (AI), thrombogenicity index (TI) and hypocholesterolemic and hypercholesterolemic FAs (H/H). Provide data in Table 3 and comment results in the discussion section. You may find equations in the following literature.

Like for proteins compare FA profile of chia seed oil with other cereals and oilseeds.

Lines 217-218 – please explain in a sentence what determines a high-quality protein

Line 221-222 – please compare phenylalanine content in chia with other plant sources and add references.

Typos and English

Line 25, 50, Table 4 – amino acid/s, please correct in the entire manuscript

Line 65 - Foss Knifetec?

Line 132 - the quantification of specific

Table 2 - 1612 kJ or 1,612 kJ

Table 3 – γ – Linolenic

Line 168 -  fibers, correct in the entire manuscript

Line 171 - particularly due to

Line 172 - high content of dietary fibers

Line 195 – not sure if edulcorant exist in English, please use a term sugar substitute

Line 200 – unsaturated

Line 213 – protein content

Line 230 – potentially

Line 239 – not sure if “intervention studies” is the correct term, please change it

Line 263 Paredes-Lopez et al (2014) [59] reported that the concentration of rosmarinic acid in chia seeds was 0.92 mg/g.

Line 279 - from Jalisco

Reviewer 2 Report

The manuscript under appreciation is about the study of nutritional and bioactive compounds of dark chia seed.

The manuscript is interesting and provides novelty.

 The following comments are to be taken into account by the authors:

Line 23: please correct “discus” to “discuss”.

Subsection 2.2: Please report in the text how did you perform the identification of fatty acids. Did you use standards?

Lines 93-95: Please specify in the text which compounds were tentatively quantified.

Lines 118-120: The calibration curve must be constructed with at least 6 points (concentrations). Please clarify in the text the number of standard concentrations.

Line 135: The phrase is not accurate. Table 2 presents the composition of chia seed and not only the results of Kjedalhl method. Please amend.

Line 140: Correct “compouunds” to “compounds” and specify in the text which compounds were tentatively identified, which databases were used, and give the literature references.

Table 4: Correct “Aminoacide” to “Amino acid”.

Table 5: Since the determination was performed in triplicates, in the column “Concentration” please provide three significant figures for the concentration and standard deviation.

Line 211: “ω-6:ω-63 ratio”   I believe the authors mean  “ω-6:ω-3 ratio”.

Reviewer 3 Report

The article is comprehensive and can be said to be one of the most complete chia seeds analyses, the data is well processed.
In the results section, the diagrams are described correctly.
The title is well described, and the text is processed according to a proper structure.
But from my point of view, the article is not new. It is complete, but in what part of science is such a detailed analysis needed? I have doubts about the practical dimension of this analysis.

Round 2

Reviewer 1 Report

Authors significantly improved their manuscript and it may be accepted.

Reviewer 2 Report

The authors have addressed all the issues. The manuscript is suitable for publication.